# Mutation bias and GC content shape antimutator invasions

Alejandro Couce [1,2] & Olivier Tenaillon[1]

Mutators represent a successful strategy in rapidly adapting asexual populations, but theory predicts their eventual extinction due to their unsustainably large deleterious load. While antimutator invasions have been documented experimentally, important discrepancies among studies remain currently unexplained. Here we show that a largely neglected factor, the mutational idiosyncrasy displayed by different mutators, can play a major role in this process. Analysing phylogenetically diverse bacteria, we find marked and systematic differences in the protein-disruptive effects of mutations caused by different mutators in species with different GC compositions. Computer simulations show that these differences can account for order-of-magnitude changes in antimutator fitness for a realistic range of parameters. Overall, our results suggest that antimutator dynamics may be highly dependent on the specific genetic, ecological and evolutionary history of a given population. This context-dependency further complicates our understanding of mutators in clinical settings, as well as their role in shaping bacterial genome size and composition.

[1] Unité Mixte de Recherche 1137 (IAME-INSERM), Université Paris Diderot, 75018 Paris, France. [2] Department of Life Sciences, Imperial College London, London SW7 2AZ, UK. Correspondence and requests for materials should be addressed to A.C. (email: acouce@imperial.ac.uk)

The idea that the mutation rate is evolvable has captivated the interest of evolutionary biologists for decades[1]. It was early recognised that, since the vast majority of mutations with phenotypic effects are deleterious, selection should primarily act to reduce the deleterious load, pushing mutation rates to be as low as physiologically affordable[2–6]. However, strains with highly-elevated mutation rates (i.e., mutators) are readily selected in clinical and laboratory populations of bacteria[7,8] and yeast[9,10], as well as in certain cancers[11]. Theory and experiments have explained this phenomenon in terms of selection pressures operating at different timescales: linkage with strong beneficial mutations enables mutators to rapidly reach fixation before their increased deleterious load becomes fully manifest, which requires the accumulation of multiple secondary deleterious mutations[12]. Due to this reliance on rapid hitchhiking, mutators are most likely to thrive whenever populations face strong selection pressures[13] and under conditions in which both recombination[14] and genetic drift[15] are unimportant.

But, eventually, populations fixed for a mutator phenotype are expected to re-evolve low mutation rates once selective pressure subsides—provided that restorative alleles are available[16,17]. Given the longer timescales involved, the evolution of reduced mutation rates has proven much more difficult to observe than its reverse process, the selection for mutator alleles. Indirect evidence comes from the fact that DNA repair genes seem to undergo frequent horizontal transfer[18–20], and the observation of marked mutation rate polymorphisms within single-patient bacterial populations[21]. Direct, empirical evidence of the evolution of reduced mutation rates is limited to a handful of experimental evolution studies[17,22–25]. The provisional picture that emerges from these studies is rather heterogeneous, with different experiments reporting contrasting findings in terms of timescales, mechanisms and magnitude of mutation rate reduction. Recent theoretical work has begun to provide a framework to account for these contrasting patterns, emphasising the role of several factors in determining the fixation probability of antimutator alleles. These factors include differences in population size, beneficial and deleterious mutation rates, mutator strength, and the availability of secondary mutations compensating the cost of deleterious mutations[12,26,27].

An additional, yet unexplored factor is the well-known mutational idiosyncrasy exhibited by different mutators[28]. This idiosyncrasy arises from the particular molecular details of the mutation-avoidance mechanism that are impaired in each mutator genotype. In *Escherichia coli*, for instance, impairment of any of the enzymes removing oxidised guanine from the DNA (e.g., MutM, MutY) results into substantial elevations of G:C → T:A mutations, while disruption of the enzyme preventing its incorporation from the free nucleotide pool (e.g., MutT) leads to a marked increase in A:T → C:G mutations[29]. These sort of mutational biases shape the tendency of different mutators to generate mutations with different fitness effects, which can have dramatic consequences on mutator success when adaptation involves just a few strongly beneficial mutations[30]. In analogy to this phenomenon, an intriguing hypothesis is that mutators that tend to generate stronger deleterious mutations may be more easily out-competed by an invading, low-mutation rate genotype. Similarly, mutators producing on average milder deleterious mutations than the wild-type may resist the invasion of antimutator alleles for longer. Whether these possibilities are plausible or not under realistic scenarios remains largely unknown.

In a first approach, at least two considerations argue against the idea that mutational spectrum differences can play any significant role in the evolution of reduced mutation rates. The first one comes from the classic Haldane-Muller principle[31,32], which states that the reduction in fitness caused by recurring deleterious mutations is roughly on the order of the deleterious mutation rate ($u_d$), irrespective of the actual fitness cost of each individual mutation ($s_d$). Such independence from $s_d$ should preclude any spectrum-driven differences in mutational load among mutators. It is well-known, however, that this principle only holds as long as $s_d > u_d$[33], a condition that may readily be violated in well-adapted, mutator populations of microbes. Second, different biases in the production of mutations are likely to translate into substantial fitness differences when just a few number of sites have a huge impact on fitness, as in the case of strongly beneficial antibiotic resistance mutations[30]. It is unclear, however, to what extent these kind of spectrum-driven differences may balance out when considering a larger number of sites. Relevant to this issue is the observation that some amino acid substitutions tend to be much more disruptive to proteins than others, a well-established fact that forms the basis of many protein alignment tools[34]. This fact affords speculation that systematic patterns may emerge at the genome-wide scale, so that different mutational spectra may produce, on average, deleterious mutations with characteristically different fitness effects.

Here, we use computer simulation to explore the extent to which the advantage of an antimutator allele deviates from the Haldane-Muller expectations under the relevant range of parameters. In addition, we estimate the genome-wide average disruptive effect on proteins of mutations caused by different mutational spectra. Importantly, since different codon usage patterns might alter the probability that a particular spectrum generates strong-effect amino acid changes, we also test whether systematic differences are to be expected among mutators in species with widely-divergent genomic GC compositions. Overall, our results suggest that mutational spectrum differences (understood as differences in the distribution of deleterious effects produced by different mutators) may play an unsuspectedly important role in the selection against high mutation rates in bacteria.

## Results

**Broad conditions allow biases to shape antimutator invasions.** To test whether mutational spectrum differences can alter the evolution of reduced mutation rates, we built a computer model that simulates the evolutionary dynamics of antimutator alleles invading a mutator population. The model was designed to capture the basic properties of the influential Lenski's Long-Term Evolution Experiment (LTEE), in which 12 *Escherichia coli* populations have been serially propagated in the same glucose-limited medium for more than 60,000 generations[35]. Crucially, one of these bacterial populations was observed to re-evolve reduced mutation rates after being dominated by a mutator phenotype for more than 10,000 generations[17]. Inspired by this experiment, we considered the simple scenario of an asexual mutator population being serially propagated in a constant environment to which is already well-adapted (see Methods). At the start of each simulation, a single antimutator allele, restoring the mutation rate to wild-type levels, is introduced. The trajectory of this allele is tracked until it either reaches fixation or is lost by drift. Multiple frequency trajectories are then used to estimate the average effective selection coefficient ($s_{eff}$) of the antimutator allele, computed empirically as the log change of the antimutator-to-mutator ratio per generation (see Methods and Supplementary Fig. 1).

Our first aim was to test whether the Haldane-Muller principle can be violated over the range of parameters typically reported in experiments with mutator bacteria. In particular, the two most important parameters for this matter are the mutation rate of mutators ($m$) and the average selection coefficient of deleterious mutations ($s_d$). Most estimates of $m$ are based on a few reporter

genes, and so caution should be exercised when using them as a proxy for genome-wide rates[36]. However, while more than a dozen genes are known to increase bacterial mutation rates when inactivated[28,37], only those causing order-of-magnitude elevations of $m$ are the ones typically observed in clinical and experimental evolution studies[37,38]. Therefore, the relevant range spans from slightly over a 10-fold increase (e.g., $mutY^-$)[39] to a 1000-fold increase of $m$ (e.g., $dnaQ^-$)[40], although all mutators observed in the LTEE fall in the several 100-fold range (e.g., $mutT^-$, $mutL^-$)[8,17].

The estimates of $s_d$ also display a certain degree of uncertainty. Attempts to estimate $s_d$ classically relied on mutation accumulation experiments, in which populations are serially passaged through single-cell bottlenecks to restrain selection from purging deleterious mutations[41]. However, since populations need to recover sufficiently after the single-cell bottleneck for the experiment to continue, there exists an upper limit on how deleterious a mutation can be to get detected, which may lead to an underestimation of $s_d$ according to the growth conditions employed[42]. Despite this limitation, different experiments have provided similar values for both the upper and the lower bounds of $s_d$. Using an early isolate from the LTEE, Kibota & Lynch[43] estimated an upper bound for $s_d$ of 0.012. Two later studies, also using *E. coli*, reported slightly higher values ($s_d$ ~0.03)[42,44]. Of note, both studies pointed to differences in mutational spectrum as a possible explanation for their higher estimates (they examined a transposon-based insertion library, and a *mutS* mutator strain, respectively). More recently, a few studies have leveraged the resolution afforded by next-generation sequencing to provide a lower bound for $s_d$. These studies reported remarkably close values for this lower bound ($s_d$ ~0.0015 to 0.0017), even though they involved three different bacterial species (*Salmonella typhimurium*;[45] *Pseudomonas aeruginosa*[46] and *Burkholderia cenocepacia*[47]). In addition, one of these studies found that $s_d$ can vary noticeably across environments[47].

Figure 1 provides a general overview of the invasion dynamics observed in the computer simulation model. In line with the Haldane-Muller expectations, we observed that the mutation rate of the resident mutator ($m$) strongly determines the speed of the antimutator invasion (Fig. 1a). However, in contrast with the Haldane-Muller principle, we found that the fitness cost of deleterious mutations ($s_d$) can also exert a substantial, albeit less dramatic effect on invasion speed (Fig. 1b). While this dependence on $s_d$ is most pronounced when mutation rates are the highest and fitness costs the smallest, our results show that invasion dynamics can indeed be affected by $s_d$ over a large fraction of the relevant range of parameters (Supplementary Fig. 2). Therefore, there are grounds to speculate whether spectrum-driven differences in $s_d$ may alter the propensity of different mutators to evolve reduced mutation rates. To examine this possibility, we expanded the computer simulation model to allow consideration of general biases in the production of deleterious mutations. We modelled these biases as a multiplicative factor ($\kappa$) that modifies the selection coefficient of deleterious mutations in the mutator background, such that when $\kappa < 1$ mutators produce milder deleterious mutations than antimutators, when $\kappa = 1$ there is no difference between backgrounds, and when $\kappa > 1$ mutations are more harmful in mutators (see Methods).

Figure 2 captures how the interplay between $s_d$ and $m$ controls the degree to which mutational spectra differences ($\kappa$) impact on the success of antimutator alleles. Two patterns can readily be appreciated by observing the overall shape of the curves in Fig. 2. First, the slopes become steeper with mutation rate ($m$) (which increases from panel $a$ to $d$). In turn, the slopes become flatter with fitness cost ($s_d$) (which increases within each panel from bottom to top). In line with the discussion in the previous

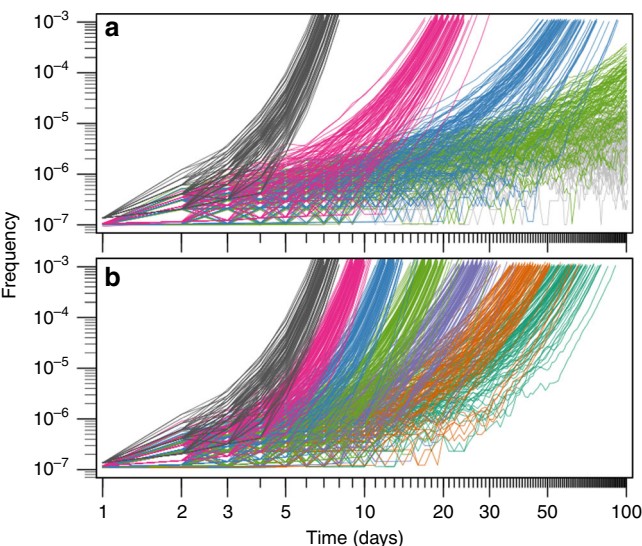

**Fig. 1** Frequency trajectories of antimutator alleles invading well-adapted, mutator populations. Lines represent 100 independent simulations for each condition. **a** Invasion dynamics under various values of the mutation rate of the resident mutator (grey, $m = 1000$; magenta, $m = 300$; blue, $m = 100$; green, $m = 30$; light grey, $m = 1$), and a fixed fitness cost of deleterious mutations ($s_d = 0.064$). **b** Invasion dynamics under various values of the fitness cost of deleterious mutations (from left to right, $s_d$ equals: 0.064, 0.032, 0.016, 0.008, 0.004, 0.002, 0.001), and a fixed mutation rate of the resident mutator ($m = 1000$). Other parameters as described in Methods

paragraph, these general patterns can be interpreted in terms of deviations from the Haldane-Muller principle. Thus, the impact of $\kappa$ is the greatest when mutation rate is maximal and fitness cost is minimal (Fig. 2d, lowest line)—exactly the same conditions under which the dependence of invasion speed on $s_d$ is most pronounced (Fig. 1b and Supplementary Fig. 2). Conversely, when populations approach the regime in which the Haldane-Muller principle holds ($s_d > u_d$), the impact of $\kappa$ becomes rather modest, which visually translates into comparatively flatter slopes (Fig. 2a, upper lines).

The previous analysis shows that the importance of mutational spectrum ultimately depends on how large the mutation rate is compared with the fitness cost of deleterious mutations. Therefore, a further natural parameter to consider is the basal deleterious mutation rate ($u_d$), that is, the absolute rate at which deleterious mutations are produced in the non-mutator background. Estimates of this quantity have classically been obtained through mutation accumulation experiments, and consequently suffer from the same uncertainties discussed for $s_d$. Throughout the previous simulations we set $u_d = 2 \times 10^{-4}$, as originally estimated in *E. coli*[43]. However, while reports in other bacteria have provided similar or slightly lower values ($u_d = 1.8-0.7 \times 10^{-4}$)[44,47], estimates in yeast differ by more than an order of magnitude, depending on whether the strain is haploid ($u_d = 1.1 \times 10^{-3}$)[48] or diploid ($u_d = 0.6-0.5 \times 10^{-4}$)[49,50]. On top of this, an additional layer of variability comes from the fact that the overall mutation rate can vary across growth conditions[51–54]. In Fig. 3a–c we explored how changes in $u_d$ within the empirically relevant range can alter the previously discussed results from Fig. 2. A prominent pattern emerging from Fig. 3 is that the slopes become steeper with larger values of $u_d$ (Fig. 3c). This result mimics the pattern found for increasing $m$ in Fig. 2, and can be understood in terms of populations moving gradually away from the Haldane-Muller regime. A more remarkable observation is that even for the lowest values tested, despite the relatively flatter slopes, the mutational

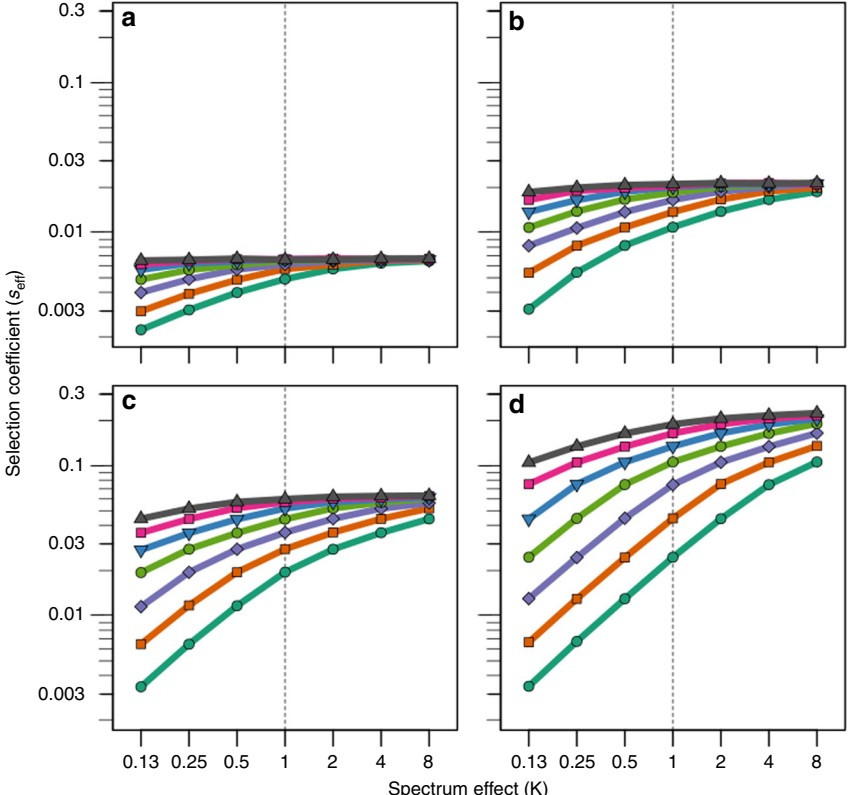

**Fig. 2** Mutational spectrum effects on the invasion speed of antimutator alleles. Points represent the effective selection coefficient ($s_{eff}$) of the invading antimutator alleles, averaged from 200 independent simulations. Panels correspond to different values of the mutation rate of the resident mutator (**a**, $m = 30$; **b**, $m = 100$; **c**, $m = 300$; **d**, $m = 1000$). Within each panel, lines depict different values for the fitness cost of deleterious mutations (from top to bottom, $s_d$ equals: 0.064, 0.032, 0.016, 0.008, 0.004, 0.002, 0.001). Mutational spectrum effects refers to the differential propensity of mutators to produce deleterious mutations with different fitness cost. We modelled this effect as a multiplicative factor ($\kappa$) that modifies $s_d$ in the mutator background, such that when $\kappa < 1$ mutators produce milder deleterious mutations than antimutators, when $\kappa = 1$ there is no difference between backgrounds, and when $\kappa > 1$ mutations are more harmful in mutators. The basal deleterious mutation rate ($u_d$) is set to $2 \times 10^{-4}$ (other parameters as described in Methods)

spectrum is still capable of exerting a moderate but sizeable impact on the performance of invading antimutator alleles.

Another issue worth considering is the lethal mutation rate ($u_l$). Lethal mutations typically occur at a much lower rate than deleterious mutations[55,56], and so as a first approximation we have neglected their influence. Lethal mutations, however, can be seen as a distinct subclass of deleterious mutations, namely, as large-effect deleterious mutations affecting essential genes. It seems possible, therefore, that mutators producing more harmful mutations may also produce a greater proportion of lethal mutations. Such spectrum-driven elevations in $u_l$, if strong enough, may alter the results discussed in Fig. 2. A further consideration is that $u_l$ is expected to be even more environmentally dependent than $u_d$, since not only the overall mutation rate varies across conditions, but also the fraction of the genome that is essential[57,58]. As a lower bound, we set $u_l = 2 \times 10^{-6}$ from estimates in *E. coli* that ~7% of the genome is unconditionally essential[58] and that ~13% of mutations within a protein are inactivating[59]. On the other hand, direct estimates in yeast have produced a value roughly an order of magnitude larger ($u_l = 3.2 \times 10^{-5}$)[60]. Figure 3d–f) confirms the intuition that lethal mutations have generally a modest effect on antimutator dynamics, except for the largest values of $\kappa$ and $u_l$ considered. How often such extreme conditions are met in natural scenarios is a matter of empirical investigation, but overall our results show that spectrum-driven variations on $u_l$ within the relevant range can play a significant, yet typically secondary role in the invasion dynamics of antimutator alleles.

We also wanted to explore to what extent the previous results can be applicable to adaptive scenarios other than the LTEE setting. In particular, we extended the simulation analyses to study the consequences of changing two key demographic parameters: the bottleneck and the maximum population size. We found that these parameters have a minor effect on antimutator dynamics even in their lower value range, in which the influence of random genetic drift begins to be noticeable (Supplementary Fig. 3). Taken together, our results support the notion that the impact of mutational spectrum on antimutator evolution can be substantial under a wide and relevant range of parameters and experimental conditions.

As a final note, it is worth highlighting that the variation in the slopes in Figs. 2 and 3 results in a large area of overlap among the curves obtained for different mutators under various combinations of parameters, especially for the smaller values of $\kappa$. This overlap represents the range of conditions under which a stronger mutator will actually be more robust to antimutator invasions than a weaker one. Since $s_d$, $u_d$ and $u_l$ can vary appreciably across species and environments, such a counterintuitive outcome illustrates the importance of considering the mutational spectrum when investigating the evolution of reduced mutation rates.

**Mutational biases cause distinct protein-disrupting patterns.** The previous results show that spectrum-driven differences in $s_d$ can greatly influence the evolution of reduced mutation rates in bacteria. It remains to be explored, nonetheless, whether

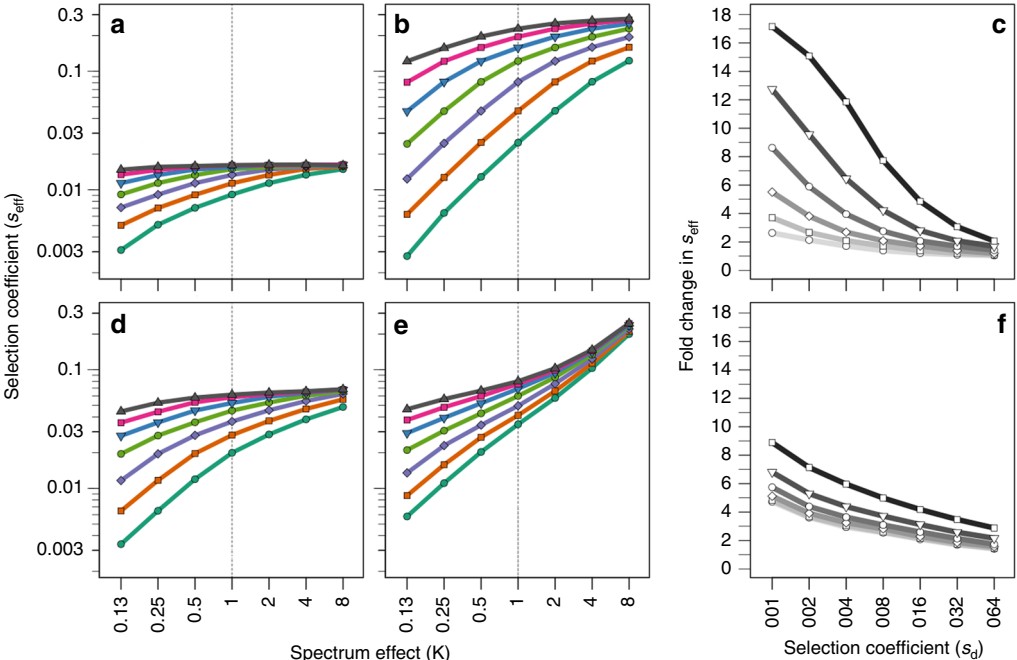

**Fig. 3** Impact of the deleterious and lethal mutation rate on antimutator dynamics. Panels **a** and **b** show antimutator fitness under two extreme values of the basal deleterious mutation rate ($u_d = 0.5 \times 10^{-4}$ and $u_d = 8 \times 10^{-4}$, respectively). Points and colours follow the same convention as in Fig. 2. Panel **c** shows the change in the antimutator's effective selection coefficient ($s_{eff}$) for various values of the basal deleterious mutation rate (from top to bottom, $u_d$ equals: $1.6 \times 10^{-3}$, $8 \times 10^{-4}$, $4 \times 10^{-4}$, $2 \times 10^{-4}$, $1 \times 10^{-4}$, $0.5 \times 10^{-4}$). Fold change refers to the change in $s_{eff}$ from $\kappa = 0.25$ to $\kappa = 4$. Panels **d** and **e** show the effects on antimutator dynamics of spectrum-driven differences in the propensity to produce lethal mutations, under two extreme values of the basal lethal mutation rate ($u_l = 0.2 \times 10^{-5}$ and $u_l = 6.4 \times 10^{-5}$, respectively). Points and colours as in Fig. 2. Panel **f** shows the change in $s_{eff}$ for various values of the basal lethal mutation rate (from top to bottom, $u_l$ equals: $6.4 \times 10^{-5}$, $3.2 \times 10^{-5}$, $1.6 \times 10^{-5}$, $0.8 \times 10^{-5}$, $0.4 \times 10^{-5}$, $0.2 \times 10^{-5}$). Fold change is defined as in **c**. In all cases, the mutation rate of the resident mutator was fixed to a single value ($m = 300$). Other parameters as described in Methods

spectrum-driven differences in $s_d$ are actually likely to occur among bacterial mutators. While differences in fitness have indeed been observed in the case of beneficial mutations involving a few genomic sites[30], the very large mutational target size for deleterious mutations may cause local, spectrum-driven differences to balance out at the genome-wide scale. However, a first look at the properties of the genetic code affords reasonable grounds for expecting the emergence of some general trends. Certainly, it has long been known that transversions are overall more detrimental than transitions, due to the fact that transversions underlie a larger fraction of non-synonymous substitutions and, among these, tend to produce changes that are less conservative of the physicochemical properties of amino acids[61–63]. Besides these trends, a closer examination reveals that the 6 types of point mutations display fairly broad distributions of disruptive effects (see Supplementary Fig. 4). Such breadth raises the possibility that, ultimately, the average disruptive effect of a given mutational spectrum may actually be determined by the highly-diverse codon usage preferences observed among bacterial species[64].

To explore these possibilities, we set out to quantify the average protein-disrupting effect of the specific point mutations elevated in 3 prominent types of mutators: $mutY^-$ (G:C → T:A), $mutT^-$ (A: T → C:G) and Mismatch Repair$^-$ (G:C → A:T, A:T → G:C) mutators[28,37] (see Methods). Briefly, we systematically computed all of the possible substitutions per codon associated with each mutational spectrum across a panel of bacterial genomes spanning a wide range of GC compositions. We then estimated the protein-disrupting effects of all these spectrum-specific substitutions by applying the well-known Grantham's matrix of physicochemical distance[65]. This amino-acid substitution matrix was previously shown to provide the best predictions of empirical

fitness effects among standard distance-based matrices[59]. As validation, we also applied an alignment-based substitution matrix (BLOSUM100)[66], which provided comparable results. Moreover, for the specific case of the LTEE experiment, we have shown that the use of Grantham's matrix provides an efficient alternative to more sophisticated and computationally intensive methods, such as Direct Coupling Analysis[57] (see Supplementary Fig. 5). Finally, seeking to increase the likelihood of non-synonymous mutations being predominantly harmful, we initially conducted these analyses for genes belonging to the COG categories most commonly enriched in essential genes (H: Coenzyme metabolism, J: Translation and M: Cell wall/membrane/envelop biogenesis)[67]—although the overall patterns remained similar when considering whole genomes (see Supplementary Fig. 6).

Figure 4 shows that there are indeed marked differences in the protein-disrupting effects of mutations caused by the different mutational spectra. The Mismatch Repair$^-$ spectrum displays the weakest disruptive effects in all tested backgrounds (Fig. 4, green), which makes sense since this spectrum comprises the two transitions, well-known to be the most conservative among all possible point mutation types[61–63]. While interesting, we shall note that this result is probably an underestimation since Mismatch Repair mutators, apart from point mutations, also exhibit an elevated occurrence of indels and large recombination events[68]. More remarkable is the fact that the disruptive effects associated with the $mutY^-$ and $mutT^-$ spectra exhibit a strong and opposite dependence on the GC content of the genetic background. In particular, we observe that the $mutY^-$ spectrum is highly detrimental in AT-rich backgrounds (Fig. 4, red), while the $mutT^-$ spectrum inflicts its greatest disruption in GT-rich backgrounds (Fig. 4, blue). This contrasting behaviour is

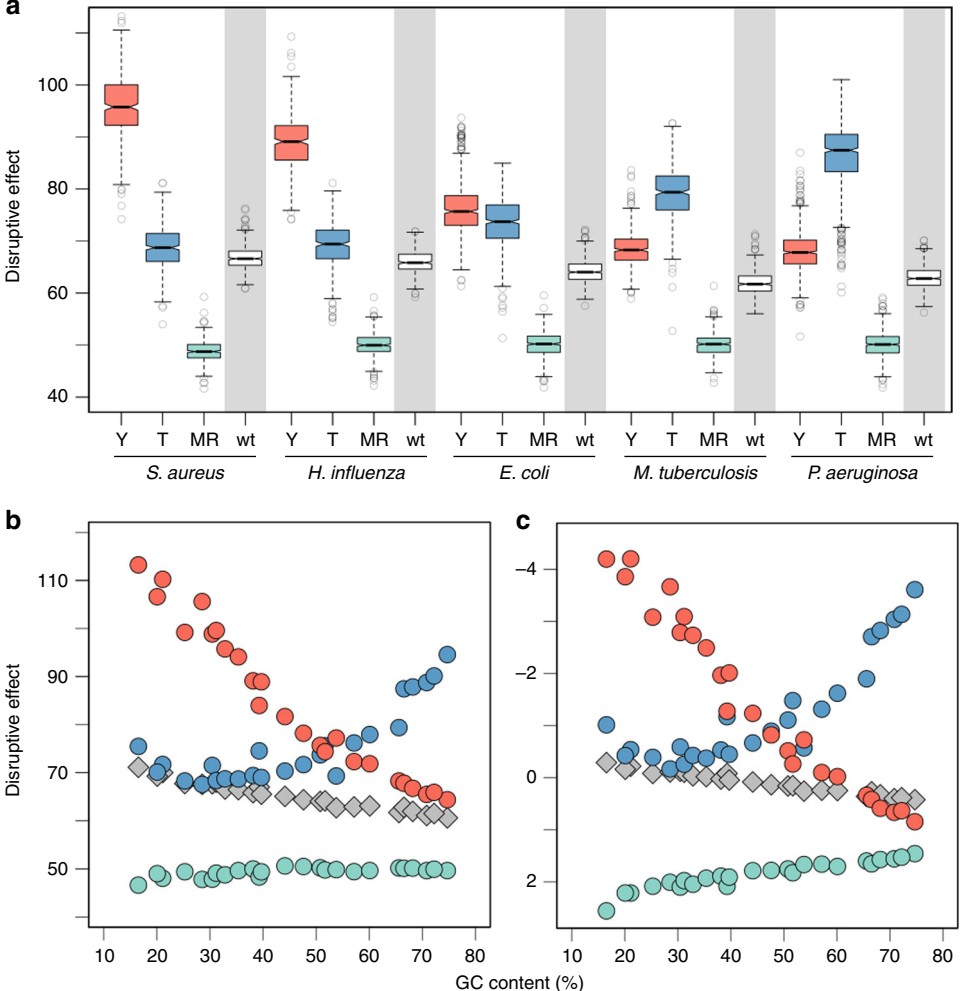

**Fig. 4** Protein-disrupting effects of mutations caused by different mutators in different genomes. Colours correspond to predictions for *mutY⁻* (red), *mutT⁻* (blue) and Mismatch Repair⁻ (green) mutators. For comparison, the effects of a unbiased spectrum are highlighted with a grey background. **a** Grantham scores in five different bacterial species for which hypermutability is of particular interest. These species are arranged, from left to right, according to increasing GC content. **b** Average Grantham scores across a panel of species with genomes spanning a wide range of GC compositions. Boxplots as defined by default. **c** Average BLOSUM100 scores across the same panel. Details about these genomes are shown in Supplementary Table 1. Source data are provided as a Source Data file

amenable to a straightforward explanation: whatever the processes causing the base composition bias may be, the last codons to be changed to conform to this bias should be the ones for which the change will produce the most harmful effects. These last codons are exactly the ones being predominantly altered by *mutY⁻*-specific mutations (G:C → T:A) and *mutT⁻*-specific mutations (A:T → C:G) in AT-rich and GT-rich backgrounds, respectively.

In addition, we should expect the fitness cost of altering these last, non-conforming codons to be the greatest in conditions where selection is weak compared to other evolutionary forces, since under such conditions selection can only prevent the most essential amino-acid sites from changing. This phenomenon would help explain why the most disruptive effects are found for the *mutY⁻* spectrum in the most AT-biased genomes—generally seen as reflective of highly-relaxed selective conditions[69–71]. This effect is better appreciated in the analyses with the distance-based instead of the aligned-based matrix (Fig. 4b versus Fig. 4c), perhaps because physicochemical distance is a more pure proxy for protein-disrupting effects than evolutionary conservation, which integrates the effects of several other factors (e.g., epistasis, basal mutational bias)[34].

## Discussion

Our analyses reveal that different mutators can be expected to produce deleterious mutations with distinctive fitness effects, and that such idiosyncrasy can greatly impact antimutator invasion dynamics. At least three points regarding these findings merit brief discussion. First, the simulations purposely focused on the effects of mutational spectra on deleterious mutations, leaving aside the complications of considering either compensatory or generally-beneficial mutations. While previous research has already studied the importance of these types of mutations on antimutator dynamics[12,26,27], a full treatment of this problem should include the fact that spectrum-driven differences can also bias mutator access to both compensatory and generally-beneficial mutations. Second, the dynamics can be further complicated by considering two phenomena well-known to limit the evolution of mutation rates: recombination and the cost of fidelity[1]. Recombination disrupts mutator hitchhiking by separating the mutator allele from its linked mutations[4,14]. Its relevance to the dynamics studied here, therefore, is probably confined to scenarios in which recombination rate is either very low or fluctuating, so as not to impede mutator fixation in the first place. Regarding the cost of fidelity, it is plausible that antimutator

alleles can exhibit differences on their direct physiological cost based on their particular genetic underpinnings (e.g., true reversion[72] versus gain-of-function mutations[16]). Characterising the complexities introduced by these factors will be reserved for future research. Third, it is worth noting the breadth of conditions under which spectrum effects are noticeable, as well as the magnitude that these effects can reach—including the paradoxical situation of weak mutators exhibiting larger deleterious load that strong mutators. The breadth and magnitude of these effects lead us to conclude that, even if taken only as a first approximation, our analyses strongly support the notion that mutational spectrum differences can greatly influence antimutator evolution in many biologically-relevant scenarios.

Finally, it is worth pointing out that the general finding of our study is that antimutator success depends not only on the extent of mutation rate elevation, but also on the mutational spectrum, the genetic background and the environmental conditions. Since the exact contribution of these factors is essentially an empirical question, it is possible that the likelihood of antimutator invasions in real-world scenarios may have to be evaluated on a case-by-case basis. Such dependence on the particulars of each case has at least two important consequences. In clinical settings, it can complicate predictions about the long-term persistence and transmissibility of mutators, thus being relevant to interventions aimed at curbing the contribution of mutators to antibiotic resistance evolution[37]. More broadly, it has implications for our views on how mutators shape the evolution of bacterial genomes. Episodes of hypermutability can be common along the evolutionary history of bacterial lineages, inflicting rapid changes in genome size and composition that can blur the signature of selection[57]. Our results suggest that the length of these pulses of hypermutability, and therefore their potential impact, may be highly dependent on the specific genetic, ecological and evolutionary history of a given lineage—a possibility further complicating the interpretation of present-day patterns of bacterial genome diversity.

## Methods

**Computer simulation.** The computer model simulates the serial passage of a bacterial population in a laboratory environment to which is already well-adapted. Since we focused on strictly asexual populations, we used a class-based model in which individuals are grouped according to their genotype[13,30]. Mimicking the serial passage protocol from the LTEE, the algorithm recreates two stages: population growth and the 1/100 bottleneck[73]. In the first stage, cells reproduce deterministically and accumulate mutations stochastically while populations expand from $10^7$ to at least $10^9$ individuals. Reproduction is formulated in terms of discrete, non-overlapping generations[74]. Every generation, individuals reproduce deterministically according to their multiplicative growth rate, defined as $r = 2 + ns_d$, where $n$ represents the number of accumulated deleterious mutations and $s_d$ is the average deleterious selection coefficient. Mutation is implemented by using a Poisson-distributed pseudorandom number generator (the function *rpois* in R). Every generation, individuals acquire deleterious mutations stochastically with a probability depending on the basal deleterious mutation rate ($u_d$) and the mutator strength ($m$) (note that for antimutator alleles $m = 1$). The second part of the algorithm is executed when population size exceeds the limit of $10^9$ individuals, and consist of taking a random sample of $10^7$ individuals, after which growth is resumed. To recover from this daily bottleneck, populations require $\geq 7$ generations (owing to discrete generation time and the accumulation of deleterious mutations).

Simulations start with a single antimutator allele entering a population of $10^7$ mutator individuals, and terminate when this allele either reaches fixation or is lost by random drift. The average effective selection coefficient of the antimutator allele is calculated empirically as $s_{eff} = log((p_g/q_g)/(p_0/q_0))/g$, where $p$ and $q$ represent the frequency of the antimutator and mutator allele, respectively, and $g$ is the number of generations[74] (see Supplementary Fig. 1). To implement the differential access of mutators to deleterious mutations with different fitness costs, we introduced a multiplicative factor ($\kappa$) that modifies $s_d$ in the mutator background as $r = 2 + \kappa ns_d$. Note that when $\kappa < 1$ mutators produce milder deleterious mutations than antimutators, when $\kappa = 1$ there are no differences between backgrounds, and when $\kappa > 1$ mutations are more harmful in the mutator background. To implement the differential propensity of mutators to produce lethal mutations, we allowed $\kappa$ to modify the basal lethal mutation rate ($u_l$) in the mutator background, such that lethal mutations represent a smaller ($\kappa < 1$), equal ($\kappa = 1$) or larger ($\kappa > 1$) than

expected proportion of the total deleterious mutations. For all tested parameter combinations, reported values of $s_{eff}$ were computed from 200 independent replicates. All programming was performed in R version 3.2.3[75], and basic codes are freely available on https://github.com/ACouce/NatComm2019.

**Genome analyses.** To conduct the bioinformatic analyses we developed a series of scripts in Python (version 2.7.12) (www.python.org). These codes were applied to a panel of 25 bacterial genomes, including relevant pathogens, and chosen to span a wide range of GC compositions. A summary of the main features of these genomes is presented in Supplementary Table 1. For all strains, the predicted coding sequences (CDSs) and their functional classification (COG) were retrieved from the Microscope platform from Genoscope (www.genoscope.cns.fr)[76]. After formatting and parsing, we estimated the average protein-disrupting effect of different mutations for all CDSs across the panel of genomes. We achieved this by computing the Grantham and BLOSUM100 scores for all of the possible substitutions per codon associated with each mutational spectrum. The Grantham and BLOSUM100 matrices were obtained from the AAindex database (www.genome.jp/aaindex)[77] and the NCBI FTP server (ftp.ncbi.nih.gov/blast/matrices), respectively. Codons harbouring incompletely specified bases (e.g., N, R, Y) were excluded from the analyses. Basic codes are freely available on https://github.com/ACouce/NatComm2019.

**Reporting summary.** Further information on research design is available in the Nature Research Reporting Summary linked to this article.

## Data availability
Genome sequences were retrieved from Genoscope and are publicly accessible at http://www.genoscope.cns.fr/agc/microscope. The source data underlying Fig. 4 and Supplementary Figs. 4, 5, 6 are provided as a Source Data file.

## Code availability
Basic codes to reproduce the results here presented are publicly available at https://github.com/ACouce/NatComm2019.

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

## Acknowledgements

We thank Dr. Harry Kemble for critical comments on an earlier version of the paper. This work was supported by the European Commission under the 7th Framework Program (ERC Grant 310944 to O.T.) and under the Horizon 2020 Framework Programme (MSCA-IF 750129 to A.C.).

## Author contributions

A.C. conceived the project, designed, conducted and interpreted simulations and genome analyses and wrote the paper. O.T. contributed to data analysis and interpretation and critically revised the paper.

## Additional information

**Competing interests:** The authors declare no competing interests.

