## [Peer Review File · Nature Communications]

Reviewers' Comments:

Reviewer #1:

Remarks to the Author:

One of the most venerable problems in evolutionary genetics concerns the evolution of mutation rate (Fisher 1930, Sturtevant 1937 and many citations thereof). Antimutator alleles, especially those that reduce it below wild-type, are central to the theory of mutation rate evolution. There is limited empirical evidence that the mutation rate is evolvable below wild-type, but it seems reasonable to conclude that antimutator alleles potentially exist, even if they're not common. Both of the prevailing theories of mutation rate evolution – the "cost of fidelity" and the "drift barrier" – predict that antimutator alleles should exist.

In this interesting paper, the authors consider the theoretical and empirical consequences of the possibility that the evolutionary fate of particular antimutator alleles may depend on the particular mutational spectrum generated by the allele, and moreover, that the fitness effects of the antimutator may depend on the genomic architecture (specifically, the GC content) of the population in which the mutator finds itself.

In general, I found this paper to be well-written and convincingly argued, almost to the point of feeling that the theoretical conclusions could hardly be otherwise. Specifically, if different types of antimutator (or mutator, for that matter) generate different mutational spectra, it stands to reason that it is at least possible that the distribution of fitness effects (DFE) generated under a particular mutational spectrum (1) may differ in general (e.g., we might expect an antimutator that reduced the frequency of indels would have a different DFE than one that reduced the frequency of base-substitutions) and (2) that the DFE of a given spectrum might depend on the population genomic context in which it finds itself (e.g., an antimutator that reduced the frequency of G-or-C to T-or-A mutations would have a different DFE if the genome was 75% GC or 25% GC).

However, the authors point out up front that there is a very powerful reason to doubt that different DFE may make any difference in antimutator dynamics, i.e., the Haldane-Muller Principle, which says that $\bar{w} = e^{-U}$, where \bar{w} is population mean fitness at mutation-selection balance and U is the genomic deleterious mutation rate. However, the authors also note that this principle holds only when the mean selective effect of a deleterious mutation (s_d) is greater than the genomic deleterious mutation rate (u_d). How widely that condition is met in nature is an empirical question, but there is reason to think that it may be violated at least sometimes. For example, if 10% of mutations are lethal and the remaining 90% have an average deleterious effect of 0.0001, $s_d \approx 5\%$, and a genomic mutation rate $>0.05/\text{generation}$ is not unrealistic.

The individual-based simulations, which are roughly parameterized from Lenski's famous experiment, are sensible, and the key result – that the mean selective effect, s_d , can influence the evolutionary dynamics of an antimutator (i.e., the H-M principle is violated) is a significant finding.

The real payoff of the paper comes in with the empirical study, in which the authors compare the putative average deleterious effects (s_d) of three different types of mutators with known differences in mutational spectrum in a diversity of bacterial genomes with known differences in GC content, with the satisfying conclusion that, yes, different mutational spectra lead to different DFEs, and the DFE is in turn dependent on the GC content.

So to sum up, this is a nice, well-written paper with clear conclusions that makes a simple but important point. The notion of the Haldane-Muller Principle is so deeply embedded in evolutionary biology that I think a clear demonstration of its limitations (and in fact one with practical implications) is indeed Nature-Worthy. I have three easy suggestions and one hard one. First, the easy ones:

1. With respect to the simulation analysis, I had a hard time keeping the intuitive meaning of the

modifying parameter kappa (κ) straight in my head. Especially given the Methods Last format, it would be worth an explicit statement that "if kappa is <1 , then (whatever) and if kappa >1 then (whatever)" (and perhaps repeated in the figure legend).

2. The term "mutational spectrum" is sometimes used interchangeably with the (implicit) term "distribution of fitness effects". As it is usually used in the population genetics literature, "mutational spectrum" has the specific meaning of the distribution of the types of molecular lesions (e.g., there are six types of base-substitutions), whereas the DFE represents the distribution of selection coefficients. Obviously, the DFE is a function of the spectrum, which is of course the main point of this paper. The attuned reader can do the translation in his/her head, but I think it would be good to explicitly draw the distinction between the two things.

3. It is, if not exactly "unconscionable", then at least it would be very nice to work in a citation of Jan Drake's classic work on antimutators.

4. Now the hard one. It would be really interesting to see how the dynamics are affected with realistic amounts of recombination. As the authors point out in the Intro, one compelling bit of evidence for the importance of antimutators is that DNA-repair genes seem to be horizontally transferred fairly often, which of course can't happen without recombination (it IS recombination). And the fact that they are implies that antimutators are relevant even with some amount of recombination. But that said, I realize that including recombination in the analysis makes things a lot more complicated, and to reiterate, this is a suggestion, not a demand.

Reviewer #2:

Remarks to the Author:

This manuscript addresses the problem of variation for mutation rate in clonal populations of bacteria. While many studies have addressed the spread of mutator clones, either theoretically or empirically, much less attention has been given to the emergence of clones with a lower mutation rate (although some recent work has been done, which the authors cite). Since mutator clones are common in both commensal and pathogenic bacteria, the question addressed in the paper is highly interesting to the community studying microbial evolution and also brings an important message in other biological context (e.g. tumor development).

The authors provide new theoretical results on the conditions by which antimutator clones could spread in populations initially composed of mutator clones. They specifically focus on a factor, mutational idiosyncrasy of the mutators, which had been previously neglected. They show that the specificities on the fitness effects of de novo mutation, associated with the fact that different types of alleles that modify the genomic mutation rate cause different mutational spectra, is an important contributor for the fate of antimutators in populations. The paper also addresses this issue amongst species of bacteria that have different patterns of codon-usage.

The work is technically sound, the paper is clearly written, although it could be improved for clarity regarding the Figure legends and methods description. The paper provides a proper account of the previous literature and its conclusions are novel to the field.

I have some major comments and a few minor.

1) Pg. 14: The description of the simulation model should be extended. In particular, it should be made very clear in the methods: i) whether the simulations are fully deterministic, ii) what is the life cycle that was implemented, in particular does random sampling only occur at the time of the periodic bottleneck?

This is critical for other researchers to be able to reproduce the work presented and understand possible limitations in the framework.

2) Line 143: What is the rationale for measuring the effective selection coefficient when the frequency increases by a 1000-fold, i.e. why 1000? and how do the results depend on this particular choice?

3) The simulation model is motivated by the Lenski LTEE propagation protocol. The authors should explain if the main results would hold true in other more general conditions, or alternatively discuss how general can one assume the conditions of the simulation model to be.

4) Fig 1. How do these invasions look like for stronger bottleneck sizes (increases intensity of genetic drift)? In addition shouldn't the figure 1a also show the case of $m=1$?

Minor:

Lines- 383-385. Clarify if population growth is deterministic or stochastic at every generation, assuming that it is a discrete generation model (but then how does the 6.7 generations period come about) and if this is an individual based simulation or based on classes of individuals grouped by the number of mutations.

Line 392- Do lethal mutations affect the results?

Line 399 – Given the definition of S_{eff} , shouldn't the Y-axis in figure 1 not be the ratio of antimutator to mutator, instead of the frequency?

The code should also be made available, which is not the case in the current github link provided by the authors.

Fig 2 legend, state what is the value of u_d .

Reviewer #1 (Remarks to the Author):

One of the most venerable problems in evolutionary genetics concerns the evolution of mutation rate (Fisher 1930, Sturtevant 1937 and many citations thereof). Antimutator alleles, especially those that reduce it below wild-type, are central to the theory of mutation rate evolution. There is limited empirical evidence that the mutation rate is evolvable below wild-type, but it seems reasonable to conclude that antimutator alleles potentially exist, even if they're not common. Both of the prevailing theories of mutation rate evolution - the "cost of fidelity" and the "drift barrier" - predict that antimutator alleles should exist.

In this interesting paper, the authors consider the theoretical and empirical consequences of the possibility that the evolutionary fate of particular antimutator alleles may depend on the particular mutational spectrum generated by the allele, and moreover, that the fitness effects of the antimutator may depend on the genomic architecture (specifically, the GC content) of the population in which the mutator finds itself.

In general, I found this paper to be well-written and convincingly argued, almost to the point of feeling that the theoretical conclusions could hardly be otherwise. Specifically, if different types of antimutator (or mutator, for that matter) generate different mutational spectra, it stands to reason that it is at least possible that the distribution of fitness effects (DFE) generated under a particular mutational spectrum (1) may differ in general (e.g., we might expect an antimutator that reduced the frequency of indels would have a different DFE than one that reduced the frequency of base-substitutions) and (2) that the DFE of a given spectrum might depend on the population genomic context in which it finds itself (e.g., an antimutator that reduced the frequency of G-or-C to T-or-A mutations would have a different DFE if the genome was 75% GC or 25% GC.

However, the authors point out up front that there is a very powerful reason to doubt that different DFE may make any difference in antimutator dynamics, i.e., the Haldane-Muller Principle, which says that $\bar{w} = e - U$, where \bar{w} is population mean fitness at mutaton-selection balance and U is the genomic deleterious mutation rate. However, the authors also note that this principle holds only when the mean selective effect of a deleterious mutation (s_d) is greater than the genomic deleterious mutation rate (u_d). How widely that condition is met in nature is an empirical question, but there is reason to think that it may be violated at least sometimes. For example, if 10% of mutations are lethal and the remaining 90% have an average deleterious effect of 0.0001, $s_d \approx 5\%$, and a genomic mutation rate $>0.05/\text{generation}$ is not unrealistic.

The individual-based simulations, which are roughly parameterized from Lenski's famous experiment, are sensible, and the key result – that the mean selective effect, s_d , can influence the evolutionary dynamics of an antimutator (i.e., the H-M principle is violated) is a significant finding. The real payoff of the paper comes in with the empirical study, in which the authors compare the putative average deleterious effects (s_d) of three different types of mutators with known differences in mutational spectrum in a diversity of bacterial genomes with known differences in GC content, with the satisfying conclusion that, yes, different mutational spectra lead to different DFEs, and the DFE is in turn dependent on the GC content.

So to sum up, this is a nice, well-written paper with clear conclusions that makes a simple but important point. The notion of the Haldane-Muller Principle is so deeply embedded in evolutionary biology that I think a clear demonstration of its limitations (and in fact one with practical implications) is indeed Nature-Worthy.

Thank you for this clear summary of our findings and their implications.

I have three easy suggestions and one hard one. First, the easy ones:

1. *With respect to the simulation analysis, I had a hard time keeping the intuitive meaning of the modifying parameter kappa (κ) straight in my head. Especially given the Methods Last format, it would be worth an explicit statement that "if kappa is <1, then (whatever) and if kappa>1 then (whatever)" (and perhaps repeated in the figure legend).*

We agree that a short reminder like the one suggested by the reviewer can improve the reader's experience. We have therefore included a short sentence in the Methods section (L435-438), and in the Figure 2 legend (L222-223).

2. *The term "mutational spectrum" is sometimes used interchangeably with the (implicit) term "distribution of fitness effects". As it is usually used in the population genetics literature, "mutational spectrum" has the specific meaning of the distribution of the types of molecular lesions (e.g., there are six types of base-substitutions), whereas the DFE represents the distribution of selection coefficients. Obviously, the DFE is a function of the spectrum, which is of course the main point of this paper. The attuned reader can do the translation in his/her head, but I think it would be good to explicitly draw the distinction between the two things.*

We agree that a sentence early on in the paper reminding the distinction between these related concepts will probably be useful for some readers. We have therefore included a short clarifying sentence in the last paragraph of the Introduction (L126).

3. *It is, if not exactly "unconscionable", then at least it would be very nice to work in a citation of Jan Drake's classic work on antimutators.*

We gladly accepted this request. Despite our best efforts to provide a fair account of the literature, several classic papers had to be left out of the manuscript due to length restrictions. We now include a citation of Drake 1993, *J Mol Evol* in the Introduction (L72) and in the Discussion sections (L383).

4. Now the hard one. It would be really interesting to see how the dynamics are affected with realistic amounts of recombination. As the authors point out in the Intro, one compelling bit of evidence for the importance of antimutators is that DNA-repair genes seem to be horizontally transferred fairly often, which of course can't happen without recombination (it IS recombination). And the fact that they are implies that antimutators are relevant even with some amount of recombination. But that said, I realize that including recombination in the analysis makes things a lot more complicated, and to reiterate, this is a suggestion, not a demand.

We agree that this is an interesting issue, and also that including recombination in our model will complicate things a great deal, probably warranting a full-blown study in its own right. It is, however, possible to say something meaningful just based on what it is already known about recombination and the evolution of mutation rate modifiers. We would argue that, although sporadic horizontal gene transfer can be an important source of antimutator alleles, the timescale at which it occurs is expected to be typically longer than the timescale of the antimutator invasions studied here. This is because, as it is well-known, recombination disrupts mutator hitchhiking by separating the mutator allele from its associated mutations (Leigh 1973, Genetics; Tenailon 2000, PNAS). Therefore, the relevance of recombination to the dynamics studied here is confined to scenarios in which recombination rate is either very low or fluctuating, so as not to impede mutator fixation in the first place.

In the revised manuscript we have added to the Discussion section a passage elaborating on this issue, its potential relevance, and the value of future work in this area (L376-384). For the sake of completeness, in this passage we also briefly raise the cost of fidelity (which the reviewer mentions above), the other phenomenon typically considered jointly with recombination in the literature on mutation rate evolution.

Reviewer #2 (Remarks to the Author):

This manuscript addresses the problem of variation for mutation rate in clonal populations of bacteria. While many studies have addressed the spread of mutator clones, either theoretically or empirically, much less attention has been given to the emergence of clones with a lower mutation rate (although some recent work has been done, which the authors cite). Since mutator clones are common in both commensal and pathogenic bacteria, the question addressed in the paper is highly interesting to the community studying microbial evolution and also brings an important message in other biological context (e.g. tumor development).

The authors provide new theoretical results on the conditions by which antimutator clones could spread in populations initially composed of mutator clones. They specifically focus on a factor, mutational idiosyncrasy of the mutators, which had been previously neglected. They show that the specificities on the fitness effects of de novo mutation, associated with the fact that different types of alleles that modify the genomic mutation rate cause different mutational spectra, is an important contributor for the fate of antimutators in populations. The paper also addresses this issue amongst species of bacteria that have different patterns of codon-usage.

The work is technically sound, the paper is clearly written, although it could be improved for clarity regarding the Figure legends and methods description. The paper provides a proper account of the previous literature and its conclusions are novel to the field.

Thank you for this clear summary of the approach, findings and relevance of our work.

I have some major comments and a few minor.

1) Pg. 14: The description of the simulation model should be extended. In particular, it should be made very clear in the methods: i) whether the simulations are fully deterministic, ii) what is the life cycle that was implemented, in particular does random sampling only occur at the time of the periodic bottleneck? This is critical for other researchers to be able to

reproduce the work presented and understand possible limitations in the framework.

We have included the requested details in the revised Methods section (L411-427). In addition, a self-contained and commented version of the code is now available at <https://github.com/ACouce/NatComm2019>.

2) Line 143: What is the rationale for measuring the effective selection coefficient when the frequency increases by a 1000-fold, i.e. why 1000? and how do the results depend on this particular choice?

We chose a 1000-fold threshold because it offered a good compromise between minimizing the noise introduced by random drift (intense when the allele is still rare) and keeping computational time reasonably short. This choice is indeed arbitrary and, as the reviewer hints at, altering it has subtle but noticeable effects: larger thresholds produce larger estimates for the effective selection coefficient. This is an unavoidable feature of trying to measure indirect selection, in which the advantage of the antimutator allele comes from its statistical association with a smaller load of deleterious mutations. Since the antimutator allele first arises on a mutator background, its advantage is initially non-existent, and it becomes manifest as mutations accumulate differentially over time. As a consequence, any threshold has to be considered as an arbitrary choice. The only 'natural' alternative is, of course, to wait until the antimutator allele reaches fixation – at the expense of much longer computational times.

In the revised version of the manuscript, we have decided to eliminate the threshold and run the simulations until the antimutator allele reaches fixation. Figures 2 and 3 have been updated accordingly. In addition, we realized that the dynamic nature of the effective selection coefficient is a technical point that is worth highlighting. For this reason, we have included a new figure in the Supplementary Material (Figure S1) that illustrates this point, together with the corresponding explanation in its caption.

3) The simulation model is motivated by the Lenski LTEE propagation

protocol. The authors should explain if the main results would hold true in other more general conditions, or alternatively discuss how general can one assume the conditions of the simulation model to be

We agree that discussing this issue will be a nice addition to the manuscript, broadening its general relevance. For this reason, we have extended the simulation analyses to explore the consequences of changing the two most critical demographic parameters: the bottleneck and the maximum population size. We focused on a relevant range for adaptive evolution experiments. In particular, we have run the simulations with either very small (1/10,000) or very large (1/2) bottleneck sizes, and either very small (10^5) or very large (10^{10}) population sizes. These explorations show that the main conclusions of our study hold true for a reasonably wide range of experimental conditions. In the revised manuscript, we have included this information by adding a new paragraph in the Results section (L263-270) and a new figure in the Supplementary Material (Figure S3).

4) Fig 1. How do these invasions look like for stronger bottleneck sizes (increases intensity of genetic drift)? In addition shouldn't the figure 1a also show the case of $m=1$?

Stronger bottlenecks cause fixation times to be shorter. This is because the bottleneck size determines the minimum frequency at which any allele (mutator or antimutator) can be present in a population, and therefore how wide the range for the antimutator-to-mutator ratio can be. This is now mentioned in the caption of the new Figure S1. The major consequence of shorter fixation times is that mutations have less time to accumulate and, therefore, the measured effective selection coefficients become proportionally smaller – as already discussed in the point 2) above. In addition, as the reviewer points out, random drift has a larger influence and increases the variability of the effective selection coefficient estimates. Similar phenomena are observed in small population sizes.

The consequences of strong bottlenecks (and small population sizes) are now illustrated in the new Supplemental Figure (Figure S3) that is also included as

part of the response to point 3). In addition, as requested, we have added the case of $m=1$ to Figure 1.

Minor:

Lines- 383-385. Clarify if population growth is deterministic or stochastic at every generation, assuming that it is a discrete generation model (but then how does the 6.7 generations period come about) and if this is an individual based simulation or based on classes of individuals grouped by the number of mutations.

Population growth is treated deterministically at every generation and, owing to discrete generation time, the number of generations needed to recover from a 1/100 bottleneck should indeed be no less than 7, and not 6.7 as originally stated. Indeed, the accumulation of deleterious mutations may in principle reduce the average growth rate of the population so that more than 7 generations elapse in each bottleneck and regrowth cycle. We have amended this passage and added the requested clarification to the Methods section (L411-427).

Line 392- Do lethal mutations affect the results?

Lethal mutations typically occur at a much lower rate than deleterious mutations, and so as a first approximation it is reasonable to assume they play a secondary role in the dynamics studied here. However, this question made us think that exploring the effects of lethal mutations will add value to the manuscript. Indeed, since lethal mutations can be seen as a distinct subclass of deleterious mutations (*i.e.*, as large-effect deleterious mutations affecting essential genes) it is possible that mutators producing more harmful mutations may also produce a greater proportion of lethal mutations. In the revised manuscript, we have explored this possibility for a realistic range of values of the basal lethal mutation rate, which is expected to vary across conditions.

The new results provide evidence that spectrum-driven elevations in the lethal mutation rate can play a significant, yet typically secondary role in the

invasion dynamics of antimutator alleles. These results are now presented and discussed in a full new paragraph added to the Results section (L245-261) and illustrated in three new panels added to the Figure 3.

Line 399 - Given the definition of $Seff$, shouldn't the Y-axis in figure 1 not be the ratio of antimutator to mutator, instead of the frequency?

We realize that plotting the ratio will probably be more appealing to the mathematically inclined reader. However, Nature Communications has a broad readership, and we think that readers from diverse backgrounds will find a frequency to be more immediately intuitive than a ratio.

We see, nonetheless, value in the suggestion by the reviewer. For this reason, we decided to show the ratio in the new figure created as part of the response to point 2) (Figure S1). Additionally, this figure is now referred to when $Seff$ is first defined in the main text (L146).

The code should also be made available, which is not the case in the current github link provided by the authors.

As stated above, a self-contained and commented version of the code is now available at <https://github.com/ACouce/NatComm2019>.

Fig 2 legend, state what is the value of ud .

Done (L224)

Reviewers' Comments:

Reviewer #1:

Remarks to the Author:

I have read the revised version of the manuscript. I was reviewer #1 for the initial submission; I liked it then and I like it even better now, given the responses to the issues raised by reviewer #2. I have only a few very minor points.

1. Spell check "mismatch" vs. "mismatch".
2. There are at least two places where a sentence or paragraph begins with "lastly" or "finally", and then the next paragraph begins with: "lastly, or "finally". Just for your consideration.
3. You may have misinterpreted my comment re the "Methods Last" with respect to where the description of kappa should be introduced. I meant that it should be fully described (i.e., $k < 1$ means mutators produce milder deleterious effects than antimutators, etc.) the FIRST place it appears in the text, i.e. L188-190 of the revision. It is good that it is in the legend to figure 2 also, but I think it would help to put the description front and center in the main text.

Reviewer #2:

Remarks to the Author:

The authors are made responded to all my comments very well. The revised manuscript now has incorporated several changes, that resolved the issues I had raised, including novel figures and their appropriate discussion.

Thanks a lot for doing this so well.

I think the paper is really important to the field of molecular evolution and also to the broader reader. I have no further comments to improve it.

Reviewers requests:

Reviewer #1 (Remarks to the Author):

I have read the revised version of the manuscript. I was reviewer #1 for the initial submission; I liked it then and I like it even better now, given the responses to the issues raised by reviewer #2. I have only a few very minor points.

Thanks again for your positive reviews and useful feedback.

1. Spell check "mismatch" vs. "mismatch".

Changes done to the proper spelling throughout the main text and the supplementary information file.

2. There are at least two places where a sentence or paragraph begins with "lastly" or "finally", and then the next paragraph begins with: "lastly, or "finally". Just for your consideration.

One occurrence of "Lastly" has now been changed to "Third" (L389 of the track-changed version), and another one has been deleted (L267 of the track-changed version).

3. You may have misinterpreted my comment re the "Methods Last" with respect to where the description of kappa should be introduced. I meant that it should be fully described (i.e., $k < 1$ means mutators produce milder deleterious effects than antimutators, etc.) the FIRST place it appears in the text, i.e. L188-190 of the revision. It is good that it is in the legend to figure 2 also, but I think it would help to put the description front and center in the main text.

Following the reviewer's advice, we now extend the description of kappa in the first place it appears in the text (L192 of the track-changed version).

Reviewer #2 (Remarks to the Author):

The authors are made responded to all my comments very well. The revised manuscript now has incorporated several changes, that resolved the issues I had raised, including novel figures and their appropriate discussion. Thanks a lot for doing this so well. I think the paper is really important to the field of molecular evolution and also to the broader reader. I have no further comments to improve it.

Thanks again for your positive reviews and useful feedback.